# Perspectives on Corrosion Inhibition Features of Novel Synthesized Gemini-Fluorinated Cationic Surfactants Bearing Varied Spacers for Acid Pickling of X60-Steel: Practical, and In Silico Calculations

**DOI:** 10.3390/ma16145192

**Published:** 2023-07-24

**Authors:** Kamal Shalabi, Hany M. Abd El-Lateef, Mohamed M. Hammouda, Amany M. A. Osman, Ahmed H. Tantawy, Mohamed A. Abo-Riya

**Affiliations:** 1Department of Chemistry, College of Science and Humanities in Al-Kharj, Prince Sattam Bin Abdulaziz University, Al-Kharj 11942, Saudi Arabia; 2Chemistry Department, Faculty of Science, Mansoura University, Mansoura 35516, Egypt; 3Department of Chemistry, College of Science, King Faisal University, Al-Ahsa 31982, Saudi Arabia; 4Department of Chemistry, Faculty of Science, Sohag University, Sohag 82524, Egypt; 5Chemistry Department, Faculty of Science, Menoufia University, Shebin El-Koam 32511, Egypt; 6Chemistry Department, Faculty of Science, Benha University, Benha 13518, Egypt

**Keywords:** fluorinated gemini surfactants, inhibition route, adsorption process, surface analysis, computational calculations

## Abstract

Through our present study, three novel Gemini-fluorinated cationic surfactants bearing different spacers (FSG6-2, FSG6-4, and FSG6-6) were synthesized, and their structures were explained via different spectroscopic instruments such as ^1^H, ^13^C, and ^19^F NMR spectra. The surface activity of the as-prepared surfactants was examined. The inhibiting influence of FSG6 molecules on the X60 steel corrosion in the pickling solution (HCl) was examined by diverse methods comprising electrochemical impedance spectroscopy (EIS), potentiodynamic polarization (PDP), and X-ray photoelectron spectroscopy (XPS) experimentations, and computational calculations. The inhibition effectiveness of FSG6 surfactants followed the order of 93.37% (FSG6-2) < 96.74% (FSG6-4) < 98.37% (FSG6-6) at 2.0 × 10^−4^ M. The FSG6 surfactants function as mixed-type inhibitors, according to PDP investigations. The H_2_O molecules that adsorbed on the steel interface were substituted with surfactant molecules, and the surfactant’s inhibitory activity is likely caused by the improvement in an adsorptive layer on the steel substrate, as specified by the EIS results. The Langmuir isotherm describes the absorption of FSG6 molecules on the metal surface. The XPS investigations validate the steel interface’s extremely protective nature. The mechanism of interaction between FSG6 molecules with an X60-steel employing the DFT calculations and MC simulations methods was also examined and discussed.

## 1. Introduction

Owing to its superior mechanical qualities and affordable price, X60-steel is a significant material that is used in a diversity of industrial applications. It is widely utilized as a building material for chemical reactors, boilers and storage tanks, heat exchanger systems, and oil and gas transport pipelines in a variety of sectors [1]. In the treatment of salt solutions, acids, and alkalis, it is also utilized in chemical and related industries.

In the industrialized sector, acid solutions are frequently employed for procedures including chemical cleaning, the acid pickling of steel, and oil-well acidification. Because it is more affordable and trouble-free than other inorganic acids, hydrochloric acid is commonly used [2,3]. The capacity of this acid to generate metal chloride, which is far more soluble in an aqueous media than sulfate, phosphate, and nitrate, gives it a significant advantage over other acids in the cleaning and pickling processes. This increase in the solubility of chloride salt leads to an increase in the corrosion rate [4,5].

The most efficient approach for defending many alloys and metals against such acid attacks is thought to be the application of corrosion inhibitors. Consequently, the investigation of steel corrosion in acid environments is of both industrial and academic interest and has attracted a lot of attention from researchers [6]. Most acid corrosion inhibitors are organic molecules comprising unsaturated bonds, electronegative atoms, for instance, nitrogen, oxygen, phosphorus, sulfur, etc., and/or aromatic rings [7,8]. It has been found that chemicals with the C-N group, polar groups, π-electrons, and electron-donating groups perform as potent inhibitors of steel in acidic media [9,10].

Fluorinated surfactants have attracted outstanding attention due to their superior surface tension-reducing activity compared to conventional surfactants [11]. The magnitude of fluorinated surfactants is greater than that of conventional ones, as wholly hydrogen atoms are replaced by fluorine atoms of significant size [12,13]. Replacing the hydrogen atoms with fluorine on the carbon backbone makes these compounds more solid and increases their hydrophobicity more than the alkyl chain [14,15]. Fluorinated surfactants diminish the surface tension more than others with the same length of hydrocarbon chain [16,17,18]. They also have a lower critical micellar concentration than their hydrocarbon counterparts, and this value decreases as the fluorocarbon chain length increases [19,20]. The superior electronegativity and reduced polarity of the fluorine atoms along the carbon chain of fluorinated surfactants impart better surface activity and increased chemical and thermal stability [21,22,23,24]. In addition, these compounds have excellent foaming, emulsification, wetting, and dispersing properties, which are present in many industrial applications, such as cosmetics, pharmaceuticals, firefighting foam, pigment additives, anti-corrosion, and food [25,26,27,28,29]. Despite these substances being widely used industrially, they are environmental hazards because they are bioaccumulating and highly toxic [30,31]. By studying the risks that fluorinated surfactants pose to the environment, it was determined that the use of compounds containing six fluorinated carbon atoms is not harmful to the environment and less toxic [32,33,34]. Consequently, fluoroalkyl chain surfactants were synthesized with a short chain even though their surface activity decreased [35,36,37]. The methods to synthesize non-toxic and environmentally safe fluorinated surfactants include decreasing the chain length, inserting a hetero atom, or using a branched-chain instead of a linear fluorinated chain [26,32,38]. Recently, an unprecedented type of surfactant called a Gemini surfactant, with two or more carbon chains with two or more head groups associated with a spacer, has been synthesized [39]. This demonstrated a superior performance and more effective surface activity compared with conventional surfactants, which gives them great importance, and they are used as models for various fields such as corrosion inhibitors, petroleum dispensing and collecting, oil recovery, nanotechnology, drug delivery, and antimicrobial agents [40,41,42,43,44]. Gemini-fluorinated surfactants were examined for their bioactivity, antifungal, and antibacterial properties. They also have strong antibacterial activity [45,46]. This article reports on the synthesis of Gemini-fluorinated surfactants with six fluorinated carbon atoms, introducing sulfur and nitrogen as hetero atoms, and as corrosion inhibitors.

Accordingly, in our study, three novel Gemini-fluorinated cationic surfactants bearing different spacers (FSG6-2, FSG6-4, and FSG6-6) were prepared and eluted via various spectroscopic techniques as ^1^H, ^13^C, and ^19^F NMR spectra. The surface activity of the prepared Gemini-fluorinated cationic surfactants was also investigated. The corrosion inhibition of the prepared fluorinated surfactants (FSG6-2, FSG6-4, and FSG6-6) was determined in an X60-steel/surfactant/pickling solution system through PDP and EIS methods. The high quality of these surfactant inhibitors depends upon their molecular construction, i.e., the number of active sites and kind of spacer existing in these surfactants. Using XPS measurements, the steel surface morphology was investigated. To ascertain the relationship between the examined fluorinated surfactants’ corrosion prevention characteristics and their molecular structures, quantum chemical calculations (i.e., DFT calculations and MC simulations) were carried out.

## 2. Materials and Methods

### 2.1. Solutions and Materials

X60-steel samples were applied to investigate the necessitated measurements, in which the metal composition in wt.% was found as follows: C, 0.23; Si, 0.38; Mn, 1.41; V, 0.06; Nb, 0.05; Ti, 0.04; P, 0.025; S, 0.015, and remainder Fe. The X60-steel samples were scratched with various SiC sheets (grades 400–1800), and acetone was used for metal rinsing, cleaned by bidistilled water, and then slightly dehydrated before soaking. Hydrochloric acid (Analar grade: 1.0 M, 37%) served as the corrosive medium.

All required chemicals and solvents, such as tetrahydrofuran, ethyl acetate, and diethyl ether, may be utilized with no need for purification and were obtained from Sinopharm Chemical Reagent Co., Ltd., Beijing, China. *N^1^,N^1^*-dimethylpropane-1,3-diamine (≥99.0%) and Perfluoro-1-hexane sulfonyl fluoride (≥98.0%) were obtained from “Aladdin Industrial corporation” company, Shanghai, China. 1,2-Diiodoethane (≥95.0%), 1,4-diiodobutane (≥97.0%), and 1,6-diiodohexane (≥99.0%) were obtained from Alfa Aesar China Co., Ltd., Shanghai, China. ^1^H (600 MHz), ^13^C (151 MHz), and ^19^F NMR (565 MHz) NMR spectra were obtained in CD_3_OD, DMSO-*d6,* or CDCl_3_ on a “Bruker AVANCE instrument” by using the inner standard as tetramethylsilane, ≥99.0%.

### 2.2. Synthetic Procedure and Characterizations

The Gemini-fluorinated cationic surfactants were synthesized in a two-step reaction.

#### 2.2.1. First Step: Preparation of N-(3-(dimethylamino)propyl)-1,1,2,2,3,3,4,4,5,5,6,6,6-tridecafluorohexane-1-sulfonamide

*N^1^,N^1^*-dimethylpropane-1,3-diamine (1.53 g, 15 mM) and (25 mM) trimethylamine in (500 mL) petroleum ether was prepared, then (6.03 g, 15 mM) perfluoro-1-hexanesulfonyl fluoride was gradually supplemented with stirring. The reaction was executed in an ice bath and stirred for 4 h at room temperature. After evaporating, the solvent C_6_F_13_SO_2_NHC_3_H_6_N(CH_3_)_2_ was obtained as a white solid with 95% yield after recrystallization in acetone. The purification of prepared compounds was achieved via washing the products with Diethyl ether to remove all unreacted materials.

##### *N*-(3-(dimethylamino)propyl)-1,1,2,2,3,3,4,4,5,5,6,6,6-tridecafluorohexane-1-sulfonamide (Appendix A)

^1^H NMR (600 MHz, DMSO-*d*_6_) (*δ*, ppm): 7.25 (s, 1H), 3.13 (s, 2H), 2.21 (s, 2H), 2.11 (s, 6H), 1.56 (s, 2H). ^13^C NMR (151 MHz, CDCl_3_) (*δ*, ppm): 159.00, 118.33, 117.42, 114.55, 114.35, 113.03, 76.21, 75.99, 75.78, 59.37, 58.97, 46.11, 45.34, 43.77, 43.19, 23.57, 23.09. ^19^F NMR (565 MHz, DMSO-*d*_6_) (*δ*, ppm): −88.43, −108.98, −119.98, −121.58, −122.58, −126.08.

#### 2.2.2. Second Step: Preparation of *N^1^,N^1^,N^2^,N^2^*-tetramethyl-*N^1^,N^2^-bis*(3-((perfluoro-alkyl)-sulfonamido)propyl)ethane-1,2-diaminium Iodide (**3a**–**c**)

A total of 100 mg, 0.17 mmol from the white powders produced in the first step was dissolved in (50 mL) tetrahydrofuran and placed in a nitrogen atmosphere. 1,2-diiodoethane, 1,4-diiodobutane, and 1,6-diiodohexane (30.01 mg, 0.081 mmol) were added to the solution. The fluorinated Gemini surfactants were attained after stirring the mixture for (8 h for 24), evaporating the reaction solvent, washing with Et_2_O, and recrystallizing in ethanol.

##### *N^1^,N^1^,N^2^,N^2^*-tetramethyl-*N^1^,N^2^-bis*(3-((perfluorohexyl)sulfonamido)propyl)ethane-1,2-diaminium Iodide (**3a**, FSG6-2) (Appendix A)

Pale brown solid, Yield: 75.1%, m.p. = 76–88 °C: ^1^H NMR (600 MHz, DMSO-*d*_6_) (*δ*, ppm): 6.87 (s, 2H), 3.59 (s, 4H), 3.46 (s, 12H), 3.11 (d, *J* = 13.6 Hz, 4H), 3.06 (s, 2H), 2.81 (s, 2H), 1.96 (t, *J* = 47.3 Hz, 2H), 1.86–1.12 (m, 2H). ^13^C NMR (151 MHz, DMSO-*d*_6_) (*δ*, ppm): 164.84, 119.44, 118.55, 118.34, 117.08, 115.99, 113.65, 65.26, 61.15, 51.62, 42.62, 23.78. ^19^F NMR (565 MHz, DMSO-*d*_6_) (*δ*, ppm): −80.28, −109.20, −119.89, −121.38, −122.85, −126.29.

##### *N^1^,N^1^,N^4^,N^4^*-tetramethyl-*N^1^,N^4^-bis*(3-((perfluorohexyl)sulfonamido)propyl)butane-1,4-diaminium Iodide (**3b**, FSG6-4) (Appendix A)

Pale yellow solid, Yield: 72.4%, m.p. = 83–86 °C: ^1^H NMR (400 MHz, DMSO-*d*_6_) (*δ*, ppm): 6.86 (s, 2H), 3.02 (s, 12H), 2.86–2.62 (m, 8H), 2.62–2.47 (m, 4H), 1.71 (s, 4H), 1.34 (d, *J* = 6.2 Hz, 4H). ^13^C NMR (151 MHz, DMSO-*d*_6_) (*δ*, ppm): 158.73, 119.44, 117.92, 115.19, 113.73, 111.21, 63.47, 60.96, 50.28, 42.89, 22.52, 17.66. ^19^F NMR (565 MHz, DMSO-*d*_6_) (*δ*, ppm): −80.37, −109.15, −119.90, −121.58, −122.58, −125.86.

##### *N^1^,N^1^,N^6^,N^6^*-tetramethyl-*N^1^,N^6^-bis*(3-((perfluorohexyl)sulfonamido)propyl)hexane-1,6-diaminium Iodide (**3c**, FSG6-6) (Appendix A)

Yellow solid, Yield: 73.8%, m.p. = 88–91 °C: ^1^H NMR (400 MHz, DMSO-*d*_6_) (*δ*, ppm): 6.86 (s, 2H), 3.32 (s, 4H), 2.82 (s, 4H), 2.62–2.45 (m, 12H), 1.90 (d, *J* = 147.1 Hz, 4H), 1.71 (t, 4H) 1.35 (d, *J* = 6.1 Hz, 4H). ^13^C NMR (151 MHz, DMSO-*d*_6_) (*δ*, ppm): 160.82, 120.38, 118.12, 115.35, 114.05, 111.27, 109.69, 63.91, 61.05, 50.36, 41.92, 31.24, 29.37, 22.50. ^19^F NMR (565 MHz, DMSO-*d*_6_) (*δ*, ppm): −80.24, −108.82, −119.89, −121.32, −122.50, −125.86.

### 2.3. Measurements of Surface Tension

Surface tension measurements were performed at 25 °C using a K6-Tensiometer (KRÜSS Co., Hamburg, Germany) with the platinum ring in distilled water. To achieve a balance, a solution of Gemini-fluorinated cationic surfactants was kept for 24 h. The platinum ring was washed more than once with distilled water and ethanol to remove any particles attached to its surface. Each reading was repeated three times before taking the final value, which was the average of these readings. Likewise, the electrical conductivity values were recorded via a conductivity analyzer (EC/TDS; type AD3000 and temperature meter). The solutions of Gemini-fluorinated cationic surfactants were prepared by water (ultrapure H_2_O) and kept for at least 24 h, and each value of electrical conductivity was the average of three measurements to minimize data errors [47].

### 2.4. Electrochemical Examinations

Electrochemical examinations were carried out using the Gamry Potentiostat/Galvanostat/ZRA device with an orthodox three-electrode organization. The counter-electrode was a platinum sheet, the working electrode was X60-steel, and the reference electrode was a silver–silver chloride (Ag/AgCl) furnished with a vitreous Luggin-capillary to diminish the impedance of the solution. All electrochemical test results were used to calculate potentials using Ag/AgCl. The X60-steel electrode was initially submerged in the test fluid for 45 min to achieve a constant open circuit voltage (*E*_OCP_). Subsequently, the EIS examination was restrained within the 100 kHz to 0. 1 Hz frequency range by means of a 10 mV irregular current signal at *E*_OCP_. ZsimpWin (v3.6) software was applied to evaluate the collected impedance data. Then, PDP tests were operated by sweeping the potential at a rate of 0.2 mV s^−1^ from +250 mV to −250 mV vs. *E*_OCP_. To minimize errors, three trials of each experiment were performed, and the temperature was thermostatically set at 50 °C.

### 2.5. Surface Characterization

The high-resolution X-ray photoelectron spectroscopy (XPS) analysis was conducted using Thermo Fisher Scientific (K-ALPHA, Waltham, MA, USA) with homochromatic X-ray Al K-alpha radiation for X60-steel samples before and after adding 2.0 × 10^−4^ M of FSG6-6 in pickling solution at 50 °C.

### 2.6. Computational Details

The Dmol3 module of the BIOVIA Materials Studio 2017 v17.1 software carried out DFT calculations with the B3LYP-functional and DNP 4.4 basis set to optimize the energy of the FSG6-3, FSG6-4 and FSG6-6 molecules in aqueous media [8]. Furthermore, the findings from DFT calculations, such as HOMO, LUMO, the energy gap (Δ*E*), electronegativity (*χ*), hardness (*η*), global softness (*σ*), and the number of electrons transferred (Δ*N*), ∆E_back-donation_ and dipole moment (*µ*), were studied and calculated using the following equations [48]:(1)$$\chi =\frac{-E_{HOMO}-E_{LUMO}}{2}$$
(2)$$\eta =\frac{1}{\sigma }=\frac{E_{LUMO}-E_{HOMO}}{2}$$
(3)$$\Delta N=\frac{\varphi -\chi_{inh}}{2(\eta_{Fe}-\eta_{inh})}$$
(4)$$\Delta N=\frac{\varphi -\chi_{inh}}{2(\eta_{Fe}-\eta_{inh})}$$
where *φ* is the function work of iron (110), χ_inh_ implies the inhibitor electronegativity, η_Fe,_ and η_inh_ are the chemical hardness of iron (0 eV) and the inhibitor, respectively.

In addition, the optimal adsorption configurations of the FSG6-2, FSG6-4, and FSG6-6 molecules on the Fe (110) surface were discovered by executing MC simulations via the adsorption locator module in BIOVIA Materials Studio 2017 program [49]. The adsorbate molecules were optimized by operating the COMPASS force field [50]. Then, the adsorption of the examined inhibitors, hydronium ions, Cl^-^ ions, and water molecules with the interface of iron (110) was completed in a model box (37.24 Å × 37.24 Å × 59.81 Å) [51].

## 3. Results and Discussion

### 3.1. Chemistry

As epitomized in Figure 1, the Gemini-fluorinated cationic surfactants were merely prepared by a two-step reaction. The reaction sequence started by reacting *N^1^,N^1^*-dimethylpropane-1,3-diamine with perfluoro-1-hexanesulfonyl fluoride in the presence of triethylamine in petroleum ether. *N-*(3-(Dimethylamino)propyl)-1,1,2,2,3,3,4,4,5,5,6,6,6-tridecafluorohexane-1-sulfonamide was obtained. The second step is the quaternization of the product acquired from the first step with 1,2-diiodoethane, 1,4-diiodobutane, and 1,6-diiodohexane in THF to generate the three Gemini-fluorinated cationic surfactants 3a-c. The yields of the products ranged from 72.4 to 75.1%, in which the structures of these products were ascertained by ^1^H-, ^13^C-, and ^19^F-NMR analyses (Appendix A).

### 3.2. Surface-Active Properties

As a special type of organic metal corrosion inhibitor, the enactment of surface-active materials is diverse compared to that of the main organic corrosion additives because this not only provides mineral protection against the corrosion process, but plays corresponding roles such as resistance corrosion, deposit resistance, and grease removal. [52].

In the current work, the surface activity in an aqueous solution of the FSG6-2, FSG6-4, and FSG6-6 was utilized to examine their behavior in the air/water interface and the probability of producing micelles within the solution. The adaptation in the relationship between surface tension (γ) and the concentration of as-prepared surfactant for FSG6-2, FSG6-4, and FSG6-6 are described and drawn in Figure 1. It was found that the early increase in the concentration of the as-prepared surfactants was investigated, causing the surface tension to quickly reduce. At a stated point, the surface tension value starts to decrease slightly, meaning that the surfactant molecules will aggregate in the solution bulk.

Based on previous studies, electrical conductivity values were used as a suitable substitute technique to measure the value of CMC. As reported in Figure 1, a study of the relationship between the surfactant’s electrical conductivity and the concentration was carried out. The figure shows two linear portions in each plot, and the critical modification of the slope refers to the CMC value of as-prepared surfactants. Subsequently, the values of CMC, defined by electrical conductivity, are more dependable, as the findings result from the surface tension, and the values are almost matching. Therefore, the following equation measures the counter-ion binding (*β*) degree [52]:(5)$$\beta =1-\frac{S_{micellar}}{S_{premicellar}}$$
where S_micellar_ is the slope of the linear line that is higher than the breakpoint, S_premicellar_ is articulated as the slope of the linear line that is lower the CMC at the k–c curve. The *β* values of as-prepared Gemini-fluorinated cationic surfactants specify the iodide ability of anti-ions to face the electrostatic force; the electrostatic force of the iodide ions in the Stern layer of the micelles may inhibit the formation of micelles [52]. Therefore, the frequent thermodynamic indices of the synthesized surfactants were calculated from the *β* values acquired from the ratio of the above slopes. As revealed in Table 1, as the hydrophobic spacer length increases, the *β* values reduce; this implies that the number of surfactant micelle aggregations decreases and the charge density over the micelle’s surface decreases [53].

Through the following calculations, other parameters, such as maximum surface effectiveness (Π_CMC_) and minimum surface-area per molecule (A_min_), maximum surface-excess concentration (Γ_max_), and the free energy of micellization/adsorption, will be calculated.

The effectiveness and efficiency of the synthesized surfactants are judged by their ability to reduce the water surface tension, Π_CMC_, defined as the efficiency (identified as the variation concerning the surface tension of distilled water *(γ*_0_) (72 mN/m at 25 °C) and surface tension at CMC *(*γ_CMC_)) of surface tension reduction, which is identified as follows:(6)$$\Pi_{CMC}=\gamma_{0}-\gamma_{CMC}$$

Table 1 shows that the γ_CMC_ values of FSG6-2, FSG6-4, and FSG6-6 are 46.63, 44.54, and 42.55 mN/m, individually, which shows that the Gemini-fluorinated cationic surfactants with a long-spacer chain can better minimize the surface tensions than the shorter spacers. The *Π_CMC_* values are found in Table 1, which shows that the lengthening of the spacer carbon chain caused an increase in the values of Π_CMC_. Besides, the as-prepared FSG6-6 has a highly effective prepared surfactant and reduces the surface tension at CMC more than other surfactants, making it close to 29.45 mN m^−1^ [53].

The superfluous surface (Γ_max_) is an applicable measurement of the process of adsorption to the air/water interface; it is known that compounds with surface activity are permanently concentrated on the surface at a higher rate than those inside the solution bulk. The main method that shows their surface activity involves the packing densities of the Gemini-fluorinated cationic surfactants at the air/aqueous solution interface. The reduced area of a single surfactant molecule (A_min_ in nm^2^) reproduces packing densities and is established by Equation (7) [54]:(7)$$\Gamma_{\max}=\frac{-1}{2.303nRT}{\left(\frac{d\gamma }{d\log C}\right)}_{T}$$
where d*γ*/dlog*C* is the slope of the linear portion of the *γ*-log *C* curve; *T* = 298 K; *R* equals 8.314 J.mol^−1^·K^−1^; Γ_max_ is mol/cm^2^, and n is a constant (equal to 2, which correlates with the number of dissolved species) [54].
(8)$$A_{\min}=\frac{10^{14}}{N_{A}\times \Gamma_{\max}}$$
where *N*_A_ equals 6.022 × 10^23^ mol^−1^; Γ_max_ and A_min_ values are listed in Table 1. The documented data show that the increase in the hydrophobicity in the spacer of the as-prepared Gemini-fluorinated cationic surfactants has a greater ability to adsorb at the interface of air/water, whereas the lengthening of the spacer leads to increases in the value of A_min_; the surfactants with a shorter hydrophobic length have a high packing density at the interface, which agrees with previous research on cationic surfactants [55].

The documented data describe that the elongation of the hydrophobic spacer leads to minimization of the Γ_max_ value; this refers to the as-prepared surfactants solutions with reduced hydrophobic spacers, which tend to adsorb at the water/air interface. The reduction in the values of Γ_max_ are based on particular factors, such as the change in the structure of the hydrophilic part, the existence of surfactants at the air/water interface, and hydrogen bonding with H_2_O to break down long-range orders of the hydrogen bonding of water molecules [56]. When the hydrophobic spacer group increases, the value of *A*_min_ also increases. This is due to Gemini-fluorinated cationic surfactants with lower hydrophobic groups that possess a greater packing density at the air/water interface. This outcome is in line with previously described surfactants [53,54,55].

To study the values of the thermodynamic restrictions of adsorption ($\Delta G_{ads}^{0}$) and micellization ($\Delta G_{mic}^{o}$) of as-prepared Gemini-fluorinated cationic surfactants, the following equations will be utilized [55]:(9)$$\Delta G_{mic}^{0}=(\beta +1)RT\ln CMC$$
(10)$$\Delta G_{ads}^{0}=(\beta +1)RT\ln CMC-\frac{6\Pi_{CMC}A_{\min}}{100}$$

Because the micellization of the as-prepared Gemini-fluorinated cationic surfactants is a class of impulsive performance and the recoded $\Delta G_{mic}^{o}$ value must be negative. Depending on the contemporary findings, it is not challenging to notice the slight difference between $\Delta G_{mic}^{o}$ and $\Delta G_{ads}^{0}$ in each cationic surfactant. The results showed that $\Delta G_{ads}^{0}$ has a negative value higher than $\Delta G_{mic}^{o}$ and a fiercer interface adsorption favorite, as a consequence of the hydrocarbon chain spacer that possesses more freedom of passage at the planar of the air/water surface between the internal fragment of the micelle [53].

### 3.3. Polarization Measurements

The corrosion of X60-Steel performance in an acidic medium in the absence and occurrence of Gemini-fluorinated cationic surfactants (FSG6-2, FSG6-4, and FSG6-6) was studied by employing the polarization diagram method, as revealed in Figure 2. It was observed that the existing density of both cathodic and anodic domains moved to inferior values in the studied Gemini surfactants (FSG6-2, FSG6-4, and FSG6-6). The assessed electrochemical strictures, such as the Tafel constants (*β*_c_ and *β*_a_), the corrosion current density (*i_cor_*), corrosion potential (*E*_corr_), protection capacity (*Z_PDP_*/%), and the surface coverage part (*θ*), are recorded in Table 2. The potentiodynamic polarization plots were attained from −250 to +250 mV concerning the *E*_OCP_ at scan rate, i.e., 1.0 mV/s [57]. The linear portions of both cathode and anode subdivisions were extrapolated to the *E*_cor_ and acquired *i*_cor_, which could be used to compute the corrosion protection capacity (*Z_PDP_*/%) of Gemini surfactants:(11)$${Ζ}_{PDP}/\%=\left(\frac{i_{cor}^{0}-i_{cor}^{i}}{i_{cor}^{0}}\right)\times 100$$
where $i_{cor}^{i}$ and $i_{cor}^{0}$ are the *i*_cor_ of the corrosive test solution containing FSG6-2, FSG6-4, and FSG6-6 and blank medium, respectively. From Table 2, the findings showed that the surfactant addition decreases both anodic steel oxidation and cathodic hydrogen evolution. The outcome of the experiment shows that when the concentrations of FSG6-2, FSG6-4, and FSG6-6 surfactants rise, the *i*_cor_ values steadily drop. Together with this, the protection ability increases with the surfactant concentration and reaches maximum values of 93.37, 96.74, and 98.37% for SG6-2, FSG6-4, and FSG6-6, respectively, at 0.2 mM. According to the approximate constant Tafel slopes of the cathodic branch *β*_c_, the surfactant reduced the surface area for the evolution of hydrogen without altering the reaction mechanism [8]. The literature survey displays that the shift in E_cor_ compared to blank provides evidence about the inhibitor type of the tested inhibitors: cathodic or anodic type when the alteration in E_cor_ is greater than 85 mV with the blank system; if not, the inhibitor is measured as mixed-type [58]. In our study, the occurrence of the surfactant in the test medium led to the displacement in the E_cor_ value being less than 85 mV, which was associated with the uninhibited system. Consequently, the examined surfactant additives might be categorized as an inhibitor of a comparatively mixed type [59]. Additionally, the electrode surface developed a protective film as a result of the decrease in corresponding i_cor_ and an increasing degree of surface covering (*θ*) with an increase inhibitor dose, confirming that the produced inhibitors operate as corrosion inhibitors. According to findings from the polarization technique, the three inhibitors’ order of inhibition capacity improves in the following sequence: FSG6-2, FSG6-4, and FSG6-6.

### 3.4. EIS Studies

To examine the processes that occur at the electrode/solution interface, the film production, and their effectiveness in terms of protection, EIS was used in pickling solution (HCl) with and without the supplement of diverse FSG6-6 concentrations, as well as the occurrence of 0.2 mM of different surfactants at 50 °C. Figure 3 shows the electrode’s Nyquist plots with and without the occurrence of surfactants.

The Bode and Bode phase modules for X60-Steel in pickling solution (HCl) with different FSG6-6 concentrations at 50 °C are seen in Figure 4A,B, after the E_OCP_ examination and with the oncoming stable-state potential. The Nyquist plots at low frequencies are best represented by a single semicircle, signifying that charge transfer monitors the corrosion process [40]. The depressed semi-circle shape of the Nyquist diagrams is frequently allied with the inhomogeneity of the X60-Steel surface that was under study in the EIS studies [47]. These depressed semicircles could be described by the X60-Steel surface’s heterogeneity or by the dispersion of some of the system’s physical property values [60]. Although the semi-circle shapes were unaffected by the presence of FSG6-2, FSG6-4, and FSG6-6 molecules, the width of the semicircles grew as the surfactant dose rose, suggesting that the corrosion reaction mechanism had not changed. As a result, the surfactant molecules must have adhered to the metal surface, and it is obvious that the adsorbed film thickness gradually increased as the surfactant concentration rose.

To find the best circuit components to match the experimental EIS results, the EIS results were fitted with a proper comparable circuit by means of the Echem Analyst software v5.6. The correctness of the simulated data was also evaluated using chi-squared. All samples obtained small chi-squares of 10^−4^, indicating that the simulated data and the real data are significantly correlated [60]. Figure 5A displays the simulated circuit [61] (R_s_(CPE-R_p_)) for the uninhibited system, which contains (i) 2-resistances: one due to the electrolyte (R_e_) and the other to the polarization (R_p_), and (ii) one constant-phase element (CPE). For an accurate and precise fit, CPE is typically used, rather than a double-layer capacitance (C_dl_). The polarization resistance, R_p_, is complemented by several types of resistance comprising the electrolyte resistance (R_e_), the charge transfer resistance (R_ct_), and film-resistance (R_f_), etc., that is, R_p_ = R_ct_ + R_f_ + R_s_. In the inhibited system demonstrated in Figure 5B, it is observed that R_p_ is in series with the parallel of capacitance (C_ads_) owing to the surfactant layer adsorption and the resistance owing to the surfactant adsorption film (R_ads_). The EIS strictures attained by fitting the experimental EIS graphs to the selected equal circuit are recorded in Table 3. *C*_dl_ is connected to the constants of CPE (*n* and *Y*_o_) by the following equation [62]:(12)$$C_{dl}={\left[Y_{0}R_{ct}^{1-n}\right]}^{\frac{1}{n}}$$

Due to a reduced dielectric constant at the interface of the metal/solution, which enables more FSG6-2, FSG6-4, and FSG6-6 molecules to adsorb on the metallic surface, surfactant-induced *C*_dl_ values are lower than those of the blank HCl solution. It is significant to observe that lower *C*_dl_ values present with a thicker adsorbed protective layer. Additionally, the decrease in *Y*_o_ as the concentration of the complex increases supports the expansion of molecules that were adsorbed onto the metallic surface. An excellent corrosion resistance is typically related to a low *C*_dl_ and high *R*_p_. Additionally, the efficiency gradually increased in response to the addition of surfactant concentrations. The highest *R*_p_ for FSG6-2, FSG6-4, and FSG6-6 molecules is 487.4, 547.3, and 725.1 cm^2^, respectively, yielding a maximum efficiency (*η*_i_) of 95.46, 95.96, and 96.95% at 2.0 × 10^−4^ M for all surfactants.

The Bode and phase angle diagrams (Figure 4B) display a distinct peak, representing the occurrence of a single time constant; as a result, the corrosion process occurs in a single phase under charge transfer controller. The phase angle also increases (toward higher negative values) and extends with the addition of the surfactant compounds, signifying good adsorption to the metallic surface, while the impedance modulus (*Z*_mod_) builds up at low frequencies [49]. A higher inhibitory capacity and a lower rate of corrosion are related to *Z*_mod_ strengthening in the low-frequency region. Additionally, the addition of the surfactant molecules shifts the Z slopes at the middle-frequency area towards −1 and 90, respectively, as a sign of increasing capacitive effectiveness [6]. The terms (*θ*) and (*η*_i_) were also estimated using Equation [55]:(13)$$\eta_{i}/\%=\theta \times 100=\left[1-\frac{R_{p}^{0}}{R_{p}^{i}}\right]\times 100$$
where $R_{p}^{0}$ and $R_{p}^{i}$ are the polarization resistances both devoid of and during the occurrence of diverse doses of FSG6-2, FSG6-4, and FSG6-6 molecules, respectively. In agreement with earlier methods, the *η*_i_/% of the assessed surfactant compounds is as follows: (FSG6-6) > (FSG6-4) > (FSG6-2), with optimal protection values of 95.46, 95.96, and 96.95%. The polarization measurements are consistent with this EIS-detected attitude.

### 3.5. Thermodynamic Parameters and Adsorption Isotherms

The surfactant adsorption on the interface of X60-steel in the pickling solution (HCl) could be a replacement process of the absorbed H_2_O molecules on the steel substrate due to the addition of surfactant compounds. For this investigation, the corrosion mitigation Equation is [63]:FSG_(sol)_ +*n*H_2_O_(ads)_ → FSG_(ads)_ + *n*H_2_O_(sol)_(14)
where FSG_(sol)_ is the FSG (surfactant) in the medium, FSG_(ads)_ is the FSG adsorbate on the steel surface, and *n* is the number of preliminary H_2_O molecules attached to the steel, replace by FSG molecule [64]. To clarify the FSG surfactants’ adsorption performance on the steel surface, diverse isotherms, for instance, Langmuir, Frumkin, Freundlich, and Temkin models, were considered, and the Langmuir isotherm model displayed the greatest fit, as given by the following equation [40]:(15)$$\frac{C_{inh}}{\theta }=\frac{1}{K_{ads}}+C_{inh}$$
where θ, C_inh_, and K_ads_ are part of the surface coverage, the surfactant dose, and the equilibrium constant of the adsorption/desorption route, respectively. The linear relation concerning C_inh_/θ and C_inh_ is exemplified in Figure 6. The regression coefficient (*R*^2^) values are detailed as 0.9988, 0.9995, and 0.9997 for FSG6-2, FSG6-4, and FSG6-6, respectively. Moreover, the slopes of the linear line were found to be 1.03, 1.01, and 0.98 for FSG6-2, FSG6-4, and FSG6-6, respectively, signifying that the surfactant additive molecules are adsorbed in a monolayer on the metal interface [47].

Furthermore, the K_ads_ values were 10.1 × 10^4^, 12.5 × 10^4^, and 15.2 × 10^4^ L/mol for FSG6-2, FSG6-4, and FSG6-6, respectively. The comparatively high value of K_ads_ inferred that FSG6-2, FSG6-4, and FSG6-6 might adsorb on the metal surface. From constant K_ads_, the adsorption-free energy $\Delta G_{ads}^{0}$ can be calculated as follows [8]:(16)$$\Delta G_{ads}^{0}=-RT\ln(55.5K_{ads})$$
where *T* is the absolute temperature, *R* = 8.314 J/mol K, and 55.5 is the molar concentration of H_2_O. The calculated $\Delta G_{ads}^{0}$ values of FSG6-2, FSG6-4, and FSG6-6 in 323 K solution are −41.7, −42.3, and −42.8 kJ mol^−1^, respectively. Compared to FSG6-2 and FSG6-4, the adsorption energy of FSG6-6 is more negative. This might demonstrate that its adsorption is more powerful. The values of $\Delta G_{ads}^{0}$ provide an indication of the attraction between the adsorbent and adsorbate, i.e., the X60-steel interface and FSG surfactant. Generally, adsorbent and adsorbate make physical contact at −20 kJ mol^−1^ or less, but a chemical interaction occurs at −40 kJ mol^−1^ or higher [49]. The values of $\Delta G_{ads}^{0}$ in our study are −41.7, −42.3, and −42.8 kJ mol^−1^ for FSG6-2, FSG6-4, and FSG6-6, respectively, indicating that the FSG surfactants interacted with the X60-steel substrate through chemisorption, which involved the interaction of FSG surfactants with the metal interface. The high inhibitory efficiency of FSG surfactants is justified by the high value of K_ads_ and $\Delta G_{ads}^{0}$, which demonstrated the robust interaction of FSG surfactants with the X60-steel interface.

### 3.6. Surface Morphology by XPS Studies

The binding of the FSG6-6 molecule with the X60-steel surface was identified using XPS analysis, proving the adsorption of FSG6-6 molecules on the X60-steel surface. The XPS spectra were found for the X60-Steel surface corroded in pickling solution without and with FSG6-6 inhibitor, and are shown in Figure 7 and Figure 8. The common peaks in C 1s, Cl 2p, Fe 2p, and O 1s were discovered for uninhibited and inhibited specimens. In addition, the adsorption of the FSG6-6 molecule on the X60-Steel surface was shown by the presence of peaks for N 1s, F 1s, I 3d, and S 2p in the inhibited specimen. The binding energies (*BE*, eV) and the respective assignment of each peak component are listed in Table 4 [65,66,67,68,69,70,71,72,73,74,75,76,77].

The de-convoluted C 1s spectra (Figure 7 and Figure 8) for the uninhibited specimen show two peaks at 285.11 and 286.86 eV, which might be assigned to−C−C− and −C−Cl bonds. Although the inhibited sample exhibits three peaks at 285.06 eV that may be ascribed to the C−C− and C−H bonds, peaks at 286.80 eV can be credited to the C−N and C−Cl bonds, and the last peak at 288.55 eV can be designated to the C−N^+^ bond [65,66]. The chlorine peak in the X60-Steel specimens treated without and with FSG6-6 in the pickling solution can be attributed to the binding of chloride ions to the positively charged Fe surface [67]. The Cl 2p spectrum exposes two approximately similar peaks (Figure 7 and Figure 8) for the specimens treated without and with FSG6-6, which are appointed to Cl 2p_3/2_ at 198.89 and 198.87 eV, and the other peaks seen for the specimens treated without and with FSG6-6 at 200.48 and 201.98 eV are attributed to Cl 2p_1/2_ [68]. The XPS spectra of Fe 2p exhibit five approximately similar peaks (Figure 7 and Figure 8) for the specimens treated without and with FSG6-6 at 711.08, 711.00 eV, designated as Fe 2p_3/2_ of Fe^2+^, 714.36, 713.20 eV and ascribed to Fe 2p_3/2_ of Fe^3+^; 718.79, 716.84 eV is attributed to Fe 2p_3/2_ satellites of Fe^2+^, 724.37, 724.64 eV are designated for Fe 2p_1/2_ of Fe^2+^, and 727.78, 72,772 eV are attributed to Fe 2p_1/2_ of Fe^3+^. Additional peaks in inhibited specimens include one at 720.00 eV, which was identified for Fe 2p_3/2_ satellites of Fe^3+^, and another at 732.90 eV, which is attibuted to Fe 2p_1/2_ satellites of Fe^3+^ [69,70]. As can be seen in Figure 7 and Figure 8, the high-resolution O 1s spectra have two peaks for the specimens treated without and with FSG6-6: the first at 530.21, 530.04 eV, referring to O^2−^ that may be bound to Fe^2+^ and Fe^3+^ in the FeO and Fe_2_O_3_ oxides [71], and the second at 531.10, 531.18 eV, which is attributed to OH^-^ that may be bound to Fe^3+^ in FeOOH [72,73].

Furthermore, the X60-Steel specimen treated with FSG6-6 in pickling solution reveals a N 1s spectrum with a single peak (Figure 8) at 400.13, which can be ascribed to the amine group (−N−H and −N−R_2_) found in the FSG6-6 molecule [74]. In addition, the F 1s spectrum has one distinctive peak (Figure 8) at 689.24 eV, due to the C−F bond present in the FSG6-6 molecule, demonstrating its adsorption on the X60-Steel surface [75]. Additionally, the I 3d spectrum with a single peak (Figure 8) at 619.63 eV may be due to the I_3_^−^ anion [76], as well as the single peak for S 2p spectrum (Figure 8) at 167.28, which is attributed to the sulphonyl (−SO_2_−) found in the FSG6-6 molecule. This demonstrates its adsorption of the X60-Steel surface [77]. Finally, the XPS analysis verifies that FSG6-6 adsorbed on the X60-Steel surface in a pickling solution.

### 3.7. Computational Calculations (DFT)

The optimized structures, LUMO, and HOMO distribution for FSG6-3, FSG6-4, and FSG6-6 molecules are depicted in Figure 9, and the linked computational adaptable is shown in Table 5. The FMO theory clarified that the LUMO and HOMO energies of an additive can identify whether donor or acceptor interactions with metal surface can occur [38]. Thus, the superior corrosion inhibitor molecule is characterized by large *E*_HOMO_ and small *E*_LUMO_ values. According to Table 5, the FSG6-6 molecule has the highest *E*_HOMO_ value of −7.09 eV, in contrast with FSG6-2 and FSG6-4 molecules (−7.29, −7.14 eV). Figure 9 shows that the HOMO level for the additive molecules is sited on the dimethylammonium and sulfonamide motifs, indicating that these positions are suitable for electrophilic assaults on the X60-Steel surface. These explanations suggest the inhibitor molecule’s proficiency for adsorption on the steel surface and, by extension, an improvement in the inhibition efficiency, which was in respectable accordance with the empirical results. However, the *E*_LUMO_ value is −2.05 eV for the FSG6-6 molecule (Table 5), less than that of the FSG6-2 and FSG6-4 molecules (−1.81, −1.92 eV). Consistent with earlier findings, the lower *E*_LUMO_ value for the FSG6-6 molecule suggests an excessive protection power for the FSG6-6 molecule.

Likewise, the Δ*E* (energy gap) is an effective factor in improving the corrosion inhibition capability of the inhibitor molecule, i.e., which enhances as the Δ*E* value is declined [78]. Table 5 shows that the FSG6-6 molecule is more likely to be adsorbed on the X60-Steel surface because its *ΔE* value (5.04 eV) is lower than those of the FSG6-2 and FSG6-4 molecules (5.48 and 5.22 eV, respectively).

Frequently, inhibitors have relatively low electronegativity (*χ*) values, suggesting their ability to contribute electrons to the metal interface [79]. On the other hand, high electronegativity (χ) values indicate that the inhibitor molecule has a great electron-withdrawing capacity, allowing for it to accept the electron from the metal surface atoms tracked by back-donation from the inhibitor molecule and form a stronger bond with the metal surface [80]. Table 5 exhibits that the electronegativity of FSG6-2, FSG6-4, and FSG6-6 molecules is somewhat higher, suggesting that these molecules can back-donate and construct a stronger bond with the X60-Steel surface.

Similarly, softness (*σ*) and hardness (*η*) can be used to evaluate the stability and reactivity of the molecule, i.e., soft molecules have a greater protection proficiency than hard molecules due to the easy relocation of electrons to the metal surface via adsorption, so they function as effective corrosion inhibitors [81]. As depicted in Table 5, FSG6-6 molecules have bigger *σ* values and slighter *η* values than FSG6-2 and FSG6-4 molecules, showing the smooth relocation of electrons to the X60-Steel substrate and their excellent inhibition properties.

Furthermore, the inhibitor’s capacity to donate or absorb electrons is determined by the fraction of electron transfer and E_back-donation_. Therefore, if the values of Δ*N* are further than zero, electron transfer is likely to occur between the inhibitor molecule and the metal interface atoms, and if the values of Δ*N* are less than zero, electron transfer can occur between the metal atoms and the inhibitor molecule (i.e., back-donation) [82]. Table 5 shows that all the *ΔN* values of the studied molecules are greater than zero, proving that the FSG6-3, FSG6-4, and FSG6-6 molecules can contribute electrons to the X60-Steel surface. When the η > 0, Δ*E*_back-donation_ is <0, indicating that back-donation occurs and the electron is withdrawn from the molecule, which is dynamically favored [83]. The negative values of E_back-donation_ for the FSG6-3, FSG6-4, and FSG6-6 molecules found in Table 5 reveal that these molecules prefer to establish strong bonds with the X60-Steel surface by back-donation [25].

The dipole moment is also a valuable feature, which is favored in the predictive pathway of corrosion inhibition [84]. The molecule’s adsorption on the metal surface is boosted, and its distortion energy is raised by the upsurge in dipole moment. Therefore, the ability to suppress corrosion is enhanced when the dipole moment increases [85]. Table 5 shows that the dipole moment value of FSG6-6 molecules is larger than that of FSG6-2 molecules (8.31 debye) and FSG6-4 molecules (12.49 debye), suggesting a greater propensity for FSG6-6 molecules to be adsorbed on the X60-Steel interface and enhance the protection.

Moreover, the molecular surface area of FSG6-2, FSG6-4, and FSG6-6 molecules has a clear correlation with their propensity to shield the X60-Steel interface in corrosive conditions. The contact area between the inhibitor molecule and the metal interface enhances as the molecular size increases, leading to a higher level of inhibitory effectiveness [86]. According to Table 5, the molecular surface area of the FSG6-6 molecule (757.18 ${Å}^{2}$) is the highest among the FSG6-2 and FSG6-4 molecules, an indication that the highest inhibition efficiency is obtained for the FSG6-6 molecule.

In addition, the Dmol^3^ module appraises molecular electrostatic potential mapping (MEP), which can reveal the active sites of inhibitor molecules. The MEP is a 3D optical component designed to demonstrate the net electrostatic significance established on a molecule from the overall charge dispensation [87]. As illustrated in Figure 10, the area with the highest electron density and, therefore, the most negative area, is represented by red in MEP maps (nucleophilic reaction); contrarily, the most positive area (electrophilic reaction) is shown by blue [88]. A visual exploration of Figure 10 confirms that the greatest negative contributions are mostly found mostly dimethyl ammonium moieties, although the electron density is lowest above tridecafluorohexane cores. The zones with superior electron density (i.e., red zones) in inhibitor species may reveal the greatest positions for interactions with the X60-Steel surface to build sturdy adsorbed protecting layers.

### 3.8. MC Simulations

The inhibitor species’ attractions to the X60-steel surface were revealed using MC simulations, and a clear concept for the adsorption mechanism was proposed. Thus, the adsorption locator module shows the optimal adsorption arrangements for FSG6-3, FSG6-4, and FSG6-6 molecules on the X60-steel surface in an acidic solution, as shown in Figure 11. This suggests an enhancement of the adsorption and the highest surface coverage [89].

More importantly, Table 6 shows the results of the MC simulations that were used to calculate the adsorption energies. It was found that the FSG6-6 molecule (−1971.88 kcal mol^−1^) has a greater negative adsorption energy value than the FSG6-2 and FSG6-4 molecules (−1848.35, −1900.63 kcal mol^−1^), suggesting that the FSG6-6 molecule energetically adsorbs on the X60-steel surface, constructing a steady adsorbed film and protecting the X60-steel from corrosion; these results are in agreement with the experimental outcomes [90]. The adsorption energy values for the FSG6-6 molecule are more negative than those for the FSG6-2 and FSG6-4 molecules, both prior to and following geometry optimization (Table 6), i.e., unrelaxed, −1662.44 kcal mol^−1^ and after, i.e., relaxed, −309.45 kcal mol^−1^, confirming that the FSG6-6 molecule has a greater protective capacity than the FSG6-2 and FSG6-4 molecules.

The d*E*_ads_/d*N*_i_ values clarify the metal/adsorbates’ arrangement energy if the adsorbed inhibitor molecule or other adsorbates molecules are excluded [49,91]. The d*E*_ads_/d*N*_i_ value for FSG6-6 molecules (−317.43 kcal mol^−1^) is superior to that of FSG6-2 and FSG6-4 molecules (−218.04, −255.39 kcal mol^−1^), as revealed in Table 6, which proves the outstanding adsorption ability of the FSG6-6 molecule compared to FSG6-2 and FSG6-4 molecules. Furthermore, the d*E*_ads_/d*N*_i_ values for water molecules, hydronium, and chloride ions are about −15.37, −52.10, and −106.35 kcal mol^−1^, respectively. These values are low relative to those of FSG6-2, FSG6-4, and FSG6-6 molecules, indicating that the adsorption inhibitor molecules are better than water molecules, hydronium ions, and chloride ions, leading to the greater dominance of inhibitor molecules compared to these other species. The combination of experimental and theoretical evidence suggests that the FSG6-2, FSG6-4, and FSG6-6 molecules adsorb resolutely on the X60-steel surface, producing a robust adsorbed defense layer that protects the X60-steel against corrosion in a pickling environment.

## 4. Conclusions

In this work, three novel fluorinated surfactants bearing different spacers (FSG6-2, FSG6-4, and FSG6-6) were synthesized and characterized. The results show that all three fluorinated surfactants have a great ability to protect X60-steel in molar HCl. Excellent corrosion resistance is typically related to a low *C*_dl_ and high *R*_p_. Additionally, the efficiency gradually rose in response to the addition of surfactant concentrations. Based on the attained findings, the protection capacities of FSG6-2, FSG6-4, and FSG6-6 followed the order 93.37% (FSG6-2) < 96.74% (FSG6-4) < 98.37% (FSG6-6) at 2.0 × 10^−4^ M, and the adsorption process of the compounds was discovered to obey the Langmuir adsorption model. The values of adsorption-free energy are −41.7, −42.3, and −42.8 kJ mol^−1^ for FSG6-2, FSG6-4, and FSG6-6, respectively, indicating that FSG surfactants interacted with the X60-steel substrate through chemisorption. Fluorinated surfactants, which are a sort of mixed-type inhibitor, decrease both the anodic and cathodic progressions according to PDP measurements. The progression of a protective barrier layer on the substrate of the steel surface due to the presence of all molecules significantly inhibits active site corrosion, as was confirmed by XPS surface morphology analysis. The electrochemical examinations were reinforced by computational studies.

## Data Availability

The raw/processed data generated in this work are available upon request from the corresponding author.

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
