# Peer review of "Perspectives on Corrosion Inhibition Features of Novel Synthesized Gemini-Fluorinated Cationic Surfactants Bearing Varied Spacers for Acid Pickling of X60-Steel: Practical, and In Silico Calculations"

_materials, 2023, doi:10.3390/ma16145192_

Round 1
Reviewer 1 Report
This is an interesting work on the use of fluorinated gemini surfactants as corrosion inhibitors. The physicochemical procedures used are described in detail. The results are thoroughly discussed on the basis of well-chosen literature.
The work is important due to the link between the effectiveness and environmental protection.
.
Author Response
REVIEWER # 1
We would like to thank the reviewer for his great efforts and giving useful criticism to the article. Below are answers to each point.
This is an interesting work on the use of fluorinated gemini surfactants as corrosion inhibitors. The physicochemical procedures used are described in detail. The results are thoroughly discussed on the basis of well-chosen literature.
The work is important due to the link between the effectiveness and environmental protection.
Author reply: Thank you for your great efforts.
Reviewer 2 Report
Reviewer Recommendation and Comments for manuscript materials-2423051y with the title: “Perspectives on corrosion inhibition features of novel synthesized Gemini fluorinated cationic surfactants bearing varied spacers for acid pickling of X60-steel: Practical, and In Silico calculations”, authors: K. Shalabi, H.M. Abd El‐Lateef, M.M. Hammouda, A.M.A. Osman, A.H. Tantawy, M.A. Abo-Riya.
The authors synthesize three perfluorinated compounds and test these compounds as corrosion inhibitors for x60 steel in hydrochloric acid solution.
The main comments that I find useful for improving the quality of the article are presented below:
*line 16* “A Through”. The typo must be corrected.
*Introduction* The typing format must be checked.
**Are the references in bold?
**English needs to be corrected/improved. For example line 47 “The less polarizing impact is produced by, the higher solubility of chloride salt, which does not slow down corrosion [4, 5].”
*line 56* “media. [5, 9].” The typo must be corrected.
*line 57* “outstanding attention due to their outstanding”. The typo must be corrected.
**What is the correct name?
N1,N1-dimethylpropane-1,3-diamine
N,N-dimethyl-1,3- propyldiamine
N,N-dimethyl-1,3- propanediamine?
*line 119* 2.2. Synthetic procedure… or 2.2. Synthesis procedure…
**what does gm mean?
**In addition to the yield, the purity of the FSG6-2/4/6 compounds should also be specified.
*line 232* “Surface active assets”?
*line 274* the order is the opposite
*line 275* “respectable capacity”?
*table 1* “mM L-1”?
**The differences between the Z% values obtained by the two electrochemical methods (EIS and PDP) (5x10-6 M) must be explained.
FSG6-2 Z%=29 (EIS) Z%=49 (PDP)
FSG6-4 Z%=43 (EIS) Z%=52 (PDP)
FSG6-6 Z%=49 (EIS) Z%=56 (PDP)
**or 323 K or 50 °C.
*3.6. Surface morphology by XPS studies*
XPS analysis shows the presence of C-C and C-Cl bonds for X60 steel in the pickling solution in the absence of inhibitor. How is that possible? In the absence of the inhibitor, these bonds cannot be observed!?
XPS analysis shows the presence of C-Cl and C=C bonds for X60 steel in the pickling solution in the presence of the inhibitor. How is that possible? If these bonds are observed, then the inhibitor is not stable in the corrosive environment. It follows that the mechanism of action/inhibition is different from that provided by the authors.
*author contribution* must be checked
*The typos must be corrected.
*The Materials journal require a specific format of references, authors must pay more attention in their writing. No reference is written according to the format required by the journal.
*There are many grammar and typing mistakes.
*The authors must revise the entire manuscript.
Extensive editing of English language required
Author Response
REVIEWER # 2
Reviewer Recommendation and Comments for manuscript materials-2423051y with the title: “Perspectives on corrosion inhibition features of novel synthesized Gemini fluorinated cationic surfactants bearing varied spacers for acid pickling of X60-steel: Practical, and In Silico calculations”, authors: K. Shalabi, H.M. Abd El‐Lateef, M.M. Hammouda, A.M.A. Osman, A.H. Tantawy, M.A. Abo-Riya.
The authors synthesize three perfluorinated compounds and test these compounds as corrosion inhibitors for x60 steel in hydrochloric acid solution.
The main comments that I find useful for improving the quality of the article are presented below:
- line 16* “A Through”. The typo must be corrected.
Author reply: Thanks for your fruitful comment. Corrected.
- Introduction* The typing format must be checked.
Author reply: Thanks for your fruitful comment. The typing format was checked.
- Are the references in bold?
Author reply: Thanks for your fruitful comment. The references weren’t in bold.
- English needs to be corrected/improved. For example line 47 “The less polarizing impact is produced by, the higher solubility of chloride salt, which does not slow down corrosion [4, 5].”
Author reply: Thanks for your fruitful comment. Corrected.
- line 56* “media. [5, 9].” The typo must be corrected.
Author reply: Thanks for your fruitful comment. Corrected.
- line 57* “outstanding attention due to their outstanding”. The typo must be corrected.
Author reply: Thanks for your fruitful comment. Corrected.
- What is the correct name?
N1,N1-dimethylpropane-1,3-diamine
N,N-dimethyl-1,3- propyldiamine
N,N-dimethyl-1,3- propanediamine?
Author reply: Thanks for your fruitful comment. Corrected as IUPAC name to N1,N1-dimethylpropane-1,3-diamine
- line 119* 2.2. Synthetic procedure… or 2.2. Synthesis procedure…
Author reply: Thanks for your fruitful comment. It is Synthetic procedure.
- what does gm mean?
Author reply: Thanks for your fruitful comment. The meaning of "gm" is abbreviation of "gram".
- In addition to the yield, the purity of the FSG6-2/4/6 compounds should also be specified.
Author reply: Thanks for your fruitful comment. The purification of prepared compounds was done via washing the products three times with Diethyl ether as a nonpolar solvent to remove all unreacted materials.
- line 232* “Surface active assets”?
Author reply: Thanks for your fruitful comment. 'Surface active assets' was changed to 'Surface-active properties'.
- line 274* the order is the opposite
Author reply: Thanks for your fruitful comment. Corrected.
- line 275* “respectable capacity”?
Author reply: Thanks for your fruitful comment. Corrected
- table 1* “mM L-1”?
Author reply: Thanks for your fruitful comment. Corrected.
- The differences between the Z% values obtained by the two electrochemical methods (EIS and PDP) (5x10-6 M) must be explained.
FSG6-2 Z%=29 (EIS) Z%=49 (PDP)
FSG6-4 Z%=43 (EIS) Z%=52 (PDP)
FSG6-6 Z%=49 (EIS) Z%=56 (PDP)
Author reply: Thanks for your fruitful comment. There is difference in Z% values obtained from PDP and EIS, but the two methods have same order of inhibition. This difference may be attributed to the current used in the technique, where PDP technique used direct current (DC) applied to the system while EIS systems characterize the time response of chemical systems using alternating current (AC) voltages over a range of frequencies.
- **or 323 K or 50 °C.
Author reply: Thanks for your fruitful comment. 50°C was unified in the manuscript.
- 3.6. Surface morphology by XPS studies*
XPS analysis shows the presence of C-C and C-Cl bonds for X60 steel in the pickling solution in the absence of inhibitor. How is that possible? In the absence of the inhibitor, these bonds cannot be observed!?
Author reply: Thanks for your fruitful comment. The C-C bond may be attributed to the presence of carbon in the carbon steel composition. While C-Cl bond may be attributed to the adsorption of Cl‑ ion on the carbon steel. https://doi.org/10.1016/j.apsusc.2019.01.149.
- XPS analysis shows the presence of C-Cl and C=C bonds for X60 steel in the pickling solution in the presence of the inhibitor. How is that possible? If these bonds are observed, then the inhibitor is not stable in the corrosive environment. It follows that the mechanism of action/inhibition is different from that provided by the authors.
Author reply: Thanks for your fruitful comment. We apologized for this mistake about the presence of C=C bonds and was corrected. While C-Cl bond may be attributed to the adsorption of Cl‑ ion on the carbon steel. https://doi.org/10.1016/j.apsusc.2019.01.149.
- author contribution* must be checked
Author reply: Thanks for your fruitful comment. The author’s contribution was checked.
- The typos must be corrected.
Author reply: Thanks for your fruitful comment. Corrected.
- The Materialsjournal require a specific format of references, authors must pay more attention in their writing. No reference is written according to the format required by the journal.
Author reply: Thanks for your fruitful comment. The Format of references was adjusted according to The Materials Style.
- There are many grammar and typing mistakes.
Author reply: Thanks for your fruitful comment. Corrected.
- The authors must revise the entire manuscript.
Author reply: Thanks for your fruitful comment. The manuscript was revised.
- Extensive editing of English language required
Author reply: Thanks for your fruitful comment. The English language was revised.

Reviewer 3 Report
Comments of materials-2423051
The main weaknesses of the manuscript:
1. The abstract can be improvised to show more significance of study.
2. Section 2.1: The specimen preparation of method and standard used to control the chemical composition and operating parameters should be provided.
3. Why the EIS examination was chosen the 100 kHz to 0.1 Hz frequency range? Why not 100 kHz to 0.01 Hz?
4. In Fig. 5, why choose model A and model B as the equivalent electrical circuit respectively, is there any references?
5. The test of neutral salt spray test should be added, because the test of corrosion resistance in the manuscript is too simple. It is not enough to only test the electrochemical performance.
6. The mechanism of synthesized Gemini fluorinated cationic surfactants bearing varied spacers for acid pickling of X60-steel did not give optimal results.
7. Please comment about the reproducibility of the images that could be acquired by this technique and statistics that corresponds to the results. The amounts of results should be expressed by mean value and standard deviation(s) in desired Tables.

Author Response
REVIEWER # 3
Comments of materials-2423051
The main weaknesses of the manuscript:
- The abstract can be improvised to show more significance of study.
Author reply: Thanks for your fruitful comment. The abstract was improved.
- Section 2.1: The specimen preparation of method and standard used to control the chemical composition and operating parameters should be provided.
Author reply: Thanks for your fruitful comment. All parameters are added.
- Why the EIS examination was chosen the 100 kHz to 0.1 Hz frequency range? Why not 100 kHz to 0.01 Hz?
Author reply: We thank the referee for this comment. We agree with the referee that many publications utilize the frequency range from 0.01 Hz to 100 kHz, but on the other hand this range is subject to be changed according to the experimental condition [1-3]. In this work we used frequency range from 0.1 Hz to 100 kHz as we have noticed that only one capacitive loop has appeared in Nyquist plots, in addition at low frequencies no new loop started to emerge. So, we think that the chosen frequency range is enough for the study and a frequency lower than 0.1 Hz will not be effective to our work. Several publications have reported the use of the same frequency range from 0.1 Hz to 100 kHz at similar experimental conditions [4-8].
[1] M. Prajila, A. Joseph, J. Mol. Liq., 241 (2017),1-8.
[2] S. Cao, D. Liu, H. Ding, J. Wang, H. Lu, J. Gui, Corros. Sci., 153 (2019), 301-313.
[3] A.A. Farag, A.S. Ismail, M.A. Migahed, Egypt. J. Pet.,27 (2018), 1187-1194.
[4] A. Popova, M. Christov, A. Vasilev, Corros. Sci, 94(2015), 70-78.
[5] A.B. Radwan, M.H. Sliem, P.C. Okonkwo, M.F. Shibl, A.M. Abdullah, J. Mol. Liq., 236 (2017), 220-231.
[6] Z. Zhang, N. Tian, L. Zhang, L. Wu, Corros. Sci., 98 (2015), 438-449.
[7] Z. Rouifi, F. Benhiba, M. El Faydy, T. Laabaissi, H. About, H. Oudda, I. Warad, A. Guenbour, B. Lakhrissi, A. Zarrouk, Chemical Data Collections, 22 (2019), 100242.
[8] N. Yilmaz, A. Fitoz, Y. Ergun, Kaan C. Emregül, Corros. Sci.,111 (2016), 110-120.
- In Fig. 5, why choose model A and model B as the equivalent electrical circuit respectively, is there any references?
Author reply: Thanks for your fruitful comment. A reference was added [61].
- The test of neutral salt spray test should be added, because the test of corrosion resistance in the manuscript is too simple. It is not enough to only test the electrochemical performance.
Author reply: Thanks for your fruitful comment. The neutral salt spray is suitable for coating while our work is corrosion inhibition.
- The mechanism of synthesized Gemini fluorinated cationic surfactants bearing varied spacers for acid pickling of X60-steel did not give optimal results.
Author reply: Thanks for your fruitful comment. the mechanism was clarified and revised.
- Please comment about the reproducibility of the images that could be acquired by this technique and statistics that corresponds to the results. The amounts of results should be expressed by mean value and standard deviation(s) in desired Tables.
Author reply: Thanks for your fruitful comment. data repeatability has been referred to in the manuscript (Highlighted in yellow) and the calculated standard deviation values have been added to the tabulated data as error values (Tables 2,3).

Round 2
Reviewer 2 Report
Reviewer Recommendation and Comments for manuscript materials-2423051 with the title: “Perspectives on corrosion inhibition features of novel synthesized Gemini fluorinated cationic surfactants bearing varied spacers for acid pickling of X60-steel: Practical, and In Silico calculations”, authors: K. Shalabi, H.M. Abd El‐Lateef, M.M. Hammouda, A.M.A. Osman, A.H. Tantawy, M.A. Abo-Riya.
The main comments that I find useful for improving the quality of the article are presented below:
Why was the Abstract highlighted in yellow if only one letter was changed? (T into t)
English needs to be corrected/improved.
SI unit for gram is g not gm?!
line 139. Why is the symbol for milligram mg and not mgm?
The differences between the Z% values obtained by the two electrochemical methods (EIS and PDP) (5x10-6 M) must be explained.
FSG6-2 Z%=29 (EIS) Z%=49 (PDP)
FSG6-4 Z%=43 (EIS) Z%=52 (PDP)
FSG6-6 Z%=49 (EIS) Z%=56 (PDP)
Author contribution must be checked
The Materials journal require a specific format of references, authors must pay more attention in their writing. No reference is written according to the format required by the journal.
The authors must revise the entire manuscript.
Moderate editing of English language required
Author Response
REVIEWER # 2
Reviewer Recommendation and Comments for manuscript materials-2423051 with the title: “Perspectives on corrosion inhibition features of novel synthesized Gemini fluorinated cationic surfactants bearing varied spacers for acid pickling of X60-steel: Practical, and In Silico calculations”, authors: K. Shalabi, H.M. Abd El‐Lateef, M.M. Hammouda, A.M.A. Osman, A.H. Tantawy, M.A. Abo-Riya.
The main comments that I find useful for improving the quality of the article are presented below:1. 1. Why was the Abstract highlighted in yellow if only one letter was changed? (T into t)
Author reply: Thanks for your fruitful comment. We apologize for this mistake.
- English needs to be corrected/improved.
Author reply: Thanks for your fruitful comment. The English was corrected/improved checked.
- SI unit for gram is g not gm?!
Author reply: Thanks for your fruitful comment. We apologize for this mistake, SI unit for gram was corrected to g.
- line 139. Why is the symbol for milligram mg and not mgm?
Author reply: Thanks for your fruitful comment. We apologize for this mistake, SI unit for gram was corrected to g.
- The differences between the Z% values obtained by the two electrochemical methods (EIS and PDP) (5x10-6 M) must be explained.
FSG6-2 Z%=29 (EIS) Z%=49 (PDP)
FSG6-4 Z%=43 (EIS) Z%=52 (PDP)
FSG6-6 Z%=49 (EIS) Z%=56 (PDP)
Author reply: Thanks for your fruitful comment. There is difference in Z% values obtained from PDP and EIS, but the two methods have same order of inhibition. This difference may be attributed to the current used in the technique, where PDP technique used direct current (DC) applied to the system while EIS systems characterize the time response of chemical systems using alternating current (AC) voltages over a range of frequencies.
- Author contribution must be checked
Author reply: Thanks for your fruitful comment. The author’s contribution was checked.
- The Materials journal require a specific format of references, authors must pay more attention in their writing. No reference is written according to the format required by the journal.
Author reply: Thanks for your fruitful comment. The Format of references was adjusted according to The Materials Style.
- The authors must revise the entire manuscript.
Author reply: Thanks for your fruitful comment. The manuscript was revised.

Reviewer 3 Report
Comments of materials-2423051
The manuscript is well revised and can be published in present form.

Author Response
REVIEWER # 3
Comments of materials-2423051
The manuscript is well revised and can be published in present form.
Author reply: Thank you for your great efforts.
